# Characterization of Wet Olive Pomace Waste as Bio Based Resource for Leather Tanning

**DOI:** 10.3390/ma14195790

**Published:** 2021-10-03

**Authors:** M. Mercè Solé, Laia Pons, Mireia Conde, Carmen Gaidau, Anna Bacardit

**Affiliations:** 1A^3^ Leather Innovation Center, Escola Politècnica Superior, Departament d’Informàtica i Enginyeria Industrial, Universitat de Lleida (UdL), 25001 Lleida, Spain; mariamerce.sole@udl.cat (M.M.S.); laia.pons@udl.cat (L.P.); mireia.conde@udl.cat (M.C.); 2INCDTP-Leather and Footwear Research Institute Division, 93, Ion Minulescu St, 3, 031215 Bucharest, Romania; carmen_gaidau@hotmail.com

**Keywords:** wet olive pomace, polyphenols, vegetable tanning, leather

## Abstract

Olive mill wastes represent an important environmental problem. Their high phenol, lipid, and organic acid concentrations turn them into phytotoxic materials. Specifically, wet olive pomace (WOP) is the waste generated in the two-phase continuous extraction process. WOP is a paste with around 60% water. The total volume of WOP generated is around 0.25 L/kg of olives processed. Its current waste management practices result in environmental problems as soil contamination, underground seepage, water-bodies pollution, and foul odor emissions. Some valorization alternatives include composting, biological treatments, direct combustion for energy production, or direct land application. The leather industry is making great efforts to apply cleaner processes while substituting chemical products for natural products. In this way, different alternatives are being studied, such as the use of zeolites, triazine derivatives, grape seed extract, olive leaf extract, etc. In this work, the use of wet olive pomace is presented as a possible alternative to conventional vegetable tannins (mimosa, quebracho, chestnut, etc.). Although different projects and studies have been developed for the valorization of olive mill wastes, there is completely a new approach to the WOP application for tanning purposes. This study shows that WOP has a significant number of polyphenolic substances, so it has a great potential to be used as a tanning agent. Specifically, this study has been able to determine that, of the polyphenols present in WOP, 39.6% correspond to tannins that are capable of tanning the skin. Additionally, it contains 14.3% non-tannins, that is, molecules that by themselves do not have the capacity to tan the leather but promote the tanning mechanism and improve the properties of the tanned leather.

## 1. Introduction

On the one hand, the leather industry transforms rawhide into leather through mechanical and chemical processes. When a ton of raw hides is converted into approximately 250 kg of leather, around 500 kg of chemicals are added during the tanning process; and between 20–50 m^3^ of wastewater are generated, between 600–900 kg of solid waste and 40 kg of emissions in the form of volatile compounds [1].

In this way, the leather industry is making great efforts to apply cleaner processes while substituting chemical products for natural products and searching for alternatives to chrome tanning such as organic tanning consisting of phenolic synthetic products, aluminum salts, as well as vegetable tanning [2,3,4,5].

Currently, the raw materials used in vegetable tanning are natural tannins obtained from different parts of plants, including woods, barks, fruits, and leaves. The most common tannins are obtained from chestnut (Castanea sativa), from quebracho wood (Schinopsis lorentzii), from Tara pods (Caesalpinia spinosa), and from the extract of mimosa obtained from the bark (Acacia mearnsii) [6].

The chemical constituents of plant tannins are made up of polymeric polyphenolic molecules. The molecules of vegetable tannins cover a wide range of molecular mass that oscillates between 300 and 5000 Da. The tanning action of polyphenols depends on the molecular mass (particle size) and the number of phenolic hydroxyl groups -OH. There is no clear theory of the fixation of plant tannins in the hide structure, although this practice has been investigated for decades. The most acceptable reaction is binding to the CO-NH bond of the protein through the phenolic hydroxyl group of the vegetable tannin, among other secondary reactions [7,8].

The terms “hydrolyzable” and “condensed” tannins are used to distinguish between the two important classes of plant tannins, that is, tannins derived from gallic acid and derived from flavan-3,4-diol, respectively. Vegetable tannins are classified according to their chemical structure: Pyrogallol or hydrolyzable tannins and Catechol or condensed tannins. Hydrolyzable tannins are divided into gallic and ellagic tannins. The most common class of condensed tannins or proanthocyanidins capable of tanning the hide are the procyanidins, which consist of polymer chains of catechin or epicatechin monomer units [9].

In conventional tanning extracts, along with tannins, there are other compounds that are separated from plant sources during extraction and concentration processes to generate commercial tannin powder. These materials, called non-tannins, are made up of carbohydrates of various types, organic acids; simple phenols that reach the molecular magnitude of tannins, salts contained in plant tissue, proteins, and lignin compounds. Among these, non-tannins are molecules that, although they cannot react with the hydroxylic groups of collagens, promote the tanning mechanism. Non-tannins can contribute to the tanning process by giving added value characteristics to the leather, such as lightfastness, which is improved by the presence of gallic acid.

On the other hand, olive mill wastes represent an important environmental problem in Mediterranean areas where they are generated in huge quantities in short periods of time. Their high phenol, lipid, and organic acid concentrations turn them into phytotoxic materials, but these wastes also contain valuable resources such as a large proportion of organic matter and a wide range of nutrients that could be recycled. Therefore, there is a need for guidelines to manage these wastes through technologies that minimize their environmental impact and lead to sustainable use of resources. Several studies have proven the negative effects of these wastes on soil microbial populations on aquatic ecosystems and even in air mediums [10].

Previous studies show that WOP has a significant number of polyphenolic substances and therefore we think they could be used as tanning agents [11,12]. Specifically, one of the most relevant in the leather industry is related to the application of the olive leaf called wetgreen^®^ OBE tanning agent, developed and patented. It is a plant-based concentrate produced from an aqueous olive leaf extract. The active tanning agents are the same ones present in some natural cosmetic articles and in extra virgin olive oil [13]. Olive oil is extracted directly from the fresh fruit of an olive tree (Olea europaea L.) using only mechanical methods, in order to maintain its natural organoleptic characteristics according to the European Commission Regulation No. 1513/2001 [14].

In an olive, there can be between 18% and 32% oil. The rest of the waste generated is vegetable water and wet olive pomace (pulp and stone). Oil extraction is carried out by centrifugation. Three different systems are normally used: the traditional batch press process (traditional mills), and continuous processes; the three-phase decanter centrifuge and the two-phase decanter centrifuge [15]. There are continuous three-phase mills, from which oil, vegetable water, and WOP come out separately, and two-phase mills (most of them), in which the vegetable water and WOP come out mixed [16].

Figure 1 is described a general vision of products, by-products, and residues in the olive oil industry [17].

The application of a novel industrial process based on the hydrothermal treatment of olive oil waste (WOP) led to a final liquid phase that contained a high concentration of simple phenolic compounds. After olive oil extraction, only a low percentage of the total phenolic compounds present in the olive fruits are found in virgin olive oil. The remaining phenolics (98–99%) end up in WOP, a by-product of the modern two-phase processing technique used in olive oil production [18]. As such, this material should be considered an important source of polyphenols [19].

WOP phenolic composition has been described by literature: catechol, 4-methylcatechol, hydroxytyrosol, tyrosol, oleuropein, p-coumaric, and other flavonoids [20]. WOP has a high-water content (up to 65%) and may contain between 2–8 g of polyphenols per kg of pomace. Other flavonoid substances can contribute to the antioxidant capability of WOP.

According to Peralbo-Molina et al., phenols identified from WOP can be classified in six different typologies: hydroxitirosol and tyrosol derivatives, iridoid precursors, secoiridoids, and derivatives, flavonoids (antioxidant activity), lignans, and phenolic acids [21].

Phenolic acids include vanillic, ferulic, gallic, or caffeic monomeric units which can contribute to confer value added to the leather when polymeric units are generated from these original monomers. Flavonoids within the molecular range of 500–3000 Da will be also an object of this study to determine their capability to interact with collagen generating hydrothermal stability to the leather.

Although different studies have been developed on the recovery of waste from the olive industry, the research presented in this article aims to contribute to a completely new approach to the application of WOP for tanning. For this reason, a characterization of the polyphenols that can be obtained in two varieties of olive as well as in different by-products obtained from olive oil extraction has been carried out: wet olive pomace (spent olives in Figure 1), degreased olive pomace, pulp, and stone (husk).

## 2. Material and Methods

A characterization of WOP composition (polyphenolic compounds, fats, tannins, and non-tannins) to assess the feasibility of using the by-products obtained from olive oil extraction as an alternative to totally/partially replace the usual tanning extracts and other synthetic tannin agents used today was performed.

Specifically, two types of olive were characterized: arbequina and palomar, and the fractions studied were: wet olive pomace, degreased olive pomace (after the second extraction), pulp, and stone.

From each of them, the following characterization was performed:-Chemical characterization: pH, moisture by gravimetry, inorganic matter, and organic matter (according to ISO 4098:2018 Determination of water-soluble matter, water-soluble inorganic matter, and water-soluble organic matter) and fats (according to ISO 4048:2018 Determination of matter soluble in dichloromethane and free fatty acid content).-Polyphenol characterization according to standard ISO 14088:2020 Quantitative analysis of tanning agents by filter method-Polyphenol characterization by high-performance liquid chromatography (HPLC-DAD)-Polyphenol characterization by HPLC-ESI/MS (Electrospray ionization, high resolution)

### 2.1. By-Products from Olive Oil Extraction

As mentioned before, two types of wet olive pomace were characterized, obtained from two different varieties of olive, grown in two Catalan towns, in 2018:

Sample 1. Palomar wet olive pomace, grown in Olesa de Montserrat (Barcelona, Spain). The olives were harvested from the tree in December 2018. They were milled in the Olesa de Montserrat mill. The virgin oil and wet olive pomace were obtained. See Figure 2.

Sample 2. Wet olive pomace that comes from Arbequina olive from Santa Maria de Miralles (Barcelona, Spain). The olives were manually harvested from the tree in December 2018. They were milled in the Agro Igualada Cooperativa mill. The virgin oil and wet olive pomace were obtained. See Figure 2.

Both wet olive pomace was kept at 4 °C in the refrigerator, with the addition of preservative Thiol S30 from KrabChemical SL (Barcelona, Spain) until use.

Additionally, three samples from General d’Olis i Derivats, S.L. (Les Borges Blanques, Spain) were characterized: degreased spent olives, pulp and stone, all of them obtained from the 2019 harvest.

Sample 3. Degreased olive pomace. It is the biomass obtained from the pomace of two or three phases once it has been partially destoned, humidity reduced in trommel-type dryers, and degreased in physical-chemical extraction plants. It is biomass that, like stone, comes from olives and is produced in large quantities in Spain. It is normally used as biomass fuel in industrial boilers for thermal and electrical production. It is the fuel used in a large part of the boilers of the agri-food industry associated with olives. In the last decade, it is exported in large quantities to Europe for fuel for industrial boilers

Sample 4. Olive pulp. It is obtained by aspiration and screening of the degrease olive pomace, being the olive pulp is a perfect protein supplement for animal feeding.

Sample 5. Olive stone. Normally it is obtained directly from the wet olive pomace or two-phase pomace through a centrifugation and screening process, it can also be obtained from the pomace by a suction and screening process. It is one of the best biomasses known, due to its high lignin content that provides a great calorific value, low ash, and good conservation. It is highly appreciated for consumption in domestic and industrial boilers and in other processes such as the production of activated carbon, etc.

### 2.2. HPLC-DAD Analysis

To determine the polyphenolic composition of the studied samples, the high-performance liquid chromatography (HPLC) method was used according to the reversed-phase separation method.

This alternative method to the official method of determination in the vegetable matter (ISO 14088:2018) that is used in the leather industry sector, allows a faster and less expensive estimation of tannin levels. It is based on the observation of the peaks obtained in the chromatograms of the analyzed samples, their identification through the spectral library, and the summation of all the areas obtained. The existing library has been completed with commercial tannin samples: mimosa, quebracho, chestnut, tara, etc., and with new polyphenolic standards like tyrosol, protocatecuic acid, hydroxytyrosol, p-cumaric acid, vanillic acid, trans-ferulic acid, catechin, epicatechin, oleuropein, and procianidine B-2. The retention time and the similarity with respect to the spectra of the identified compounds calculated by a spectral comparison algorithm have been taken into account.

All types of tannins have strong absorption of ultraviolet light since they are constituted by the polymerization of polyphenols. To carry out the qualitative determination of tannins in plant extracts by means of HPLC, it must be taken into account that tannins share the same chromophore group. In the case of hydrolyzable tannins, it is gallic acid and in the case of catechinic tannins, it is catechin and to lesser extent epicatechin.

In this study, different polyphenolic patterns of the catechinic and hydrolyzable type (galotanins and ellagitannins) have also been incorporated into the existing library.

The chromophores of polyphenols of the catechinic, galotanin, or elagitanin type are different from each other, a characteristic that allows the classification of each polyphenol in the sample into one of these groups, even though its structure has not been completely identified nor is its molecular weight known. See Figure 3.

In this way, knowing that all tannins share the same chromophore group, we can have molecules of very different molecular weight, but which have practically the same UV-Vis spectrum. Small differences in configuration will slightly modify peak wavelengths and molar absorptivity values but will always follow the same pattern with respect to the spectrum.

The equipment used is an HPLC Alliance from Waters (Barcelona, Spain). An XBridge Phenyl separation column (3.5-micron particle size, 15 cm length, 130 Å pore diameter from Waters) is used with a PDA detector between 200 and 400 nm. Specifically, at 271.1 nm. XBridge Phenyl columns have applicability in separations where alternative selectivity is needed, especially as the analytes of interest contain an aromatic ring. These columns offer alternative selectivity over straight-chain alkyl columns, providing great flexibility in difficult-to-resolve separations.

The mobile phase was composed of eluent A (ultrapure water acidified with 0.1% formic acid) and eluent B (acetonitrile acidified with 0.1% formic acid). The reverse-phase working method is constituted by a gradient of eluents according to the following work program: 95%A-5%B from 0 to 6 min, 74% A-26% B, from 6 to 30 min, 0% from A-100% B from 30 to 34 min. The mobile phase flow is 1 mL/min. The working temperature of the column was set at 35 °C.

All the reagents used are of the necessary quality for HPLC assays. Millipore quality ultrapure water is used for the preparation of the solutions. All solutions are filtered through 0.45μm nylon filters prior to HPLC-DAD analysis.

### 2.3. HPLC-ESI/MS Analysis

To complete the characterization, the olive pulp was studied by means of the HPLC-ESI/MS analysis. To this purpose, an aqueous solution of olive pulp from 1 to 25 is prepared in ultrapure water. The solution is homogenized using vortex for 10 min and filtered with filter paper and later with a 0.45 μm syringe filter.

The methodology used is mass spectrometry coupled to LC / MS high-resolution liquid chromatography (high-resolution ESI). The LC / MSD-TOF instrument used is a mass spectrometer with a time-of-flight analyzer.

The mass technique (MS) used in the sample test corresponds to the Electrospray (Ion Spray) (ESI-MS).

The positive and negative ions are read. The detector consists of a source with a double nebulizer, which allows the simultaneous introduction of internal reference by means of an independent nebulizer, for the measurement of exact masses.

The working conditions in the mass spectrometer were as follows: capillary 4000 V (+), 3500 V (−), drying Gas 10.0 L/min, nebulizer pressure 30 Psi, gas temperature 325 °C, fragmenter 175 V (+), and scan 105–2000 m/z.

To analyze the degree of response of the sample and optimize the values of the working parameters, the previously prepared solution of the sample was carried out by direct introduction (ESI (+ and −)).

Ten microliters of the sample solution prepared above were injected into the spectrometer through an Agilent 1100 HPLC system, using a mixture of water: Acetonitrile 1:1 and 0.1% formic acid as eluent, applying a flow of 220 µL/min.

### 2.4. Lab Scale Application

In order to assess the wet olive pomace capacity of tanning, it was applied on leather, on pickled hide, following the formulation shown in Table 1.

## 3. Results and Discussion

As mentioned before, the aim of this study is to assess the feasibility of using the by-products obtained from olive oil extraction as an alternative to totally/partially replace the usual tanning extracts and other synthetic tannin agents used currently.

### 3.1. Chemical Characterization

The first stage consisted of determining the chemical characterization and content of the polyphenols of the 5 samples under study in order to choose the most suitable fraction to be able to tan leather. The results obtained are shown in Table 2.

As mentioned before, in the leather industry sector, the number of polyphenols able to tan leather are determined according to standard ISO 14088:2020 Quantitative analysis of tanning agents by filter method that allows the quantification of tannin content, no tannins, non-soluble substances, soluble substances, and solids by gravimetry. The results can be seen in Table 3.

As can be seen in Table 2 and Table 3, Palomar and Arbequina WOP contain 10.7% of fats and 7.8%, respectively, and contain between 2.6–3.0% tannins, 66% moisture, and 28% non-soluble substances.

The degreased olive pomace contains 3.7% tannins, 83.1% non-soluble substances, and 3.4% fat. It contains 6.25% inorganic matter, 3% more than the inorganic matter in WOP.

The olive pulp contains 3.5% tannins, 77.2% non-soluble substances, 4.2% fat, and 6.1% inorganic matter. As it contains almost no olive stone, it reduces the content of non-soluble solids.

The olive stone contains almost no tannins, just 0.7%. It contains many non-soluble substances, specifically 89.9% and 0.74% of inorganic matter.

As can be seen in the results obtained, each type of olive and each type of fraction present a different amount of fat and tannins. The difference in fats they present, apart from the olive species, is also due to the type of mill used, which can have different yields due to differences in working temperature and grinding pressure.

In this stage of the study, the olive stone is discarded due to the low tannin content and the high non-soluble substances content. For this reason, the olive stone is rejected as a matter to tan leather.

### 3.2. HPLC-DAD Characterization

Once the first quantification of polyphenols able to tan was obtained in the different fractions studied, a semi-quantification by HPLC-DAD was carried out in order to identify the types (i.e., catechin, epicatechin, gallic, ellagic, etc.).

As mentioned before, a comparison of the algorithms of the UV spectra of the peaks of the chromatograms with those of the digital library was made. Each peak was individually reviewed, checking the coincidence of its retention time and UV spectrum with the polyphenols of the spectra recorded in the library. Once the results were obtained, the sum of the total area of the peaks identified in the chromatogram as non-tannins, tannins, and unidentified was performed.

Table 4 shows a summary of the percentage in tannins, non-tannins, and unidentified obtained by each of the fractions studied.

The results by HPLC analysis show the same conclusions as the results obtained following the Standard ISO 14088: 2020 Quantitative analysis of tanning agents by filter method. The results show that the fraction that contains the most amount of tannins, and of which more types can be identified according to the database of the equipment used, is olive pulp.

In Figure 4 the chromatogram obtained from olive pulp can be seen. In addition, in Table 5 the types of polyphenols that have been identified are detailed.

As can be seen in Table 5, in the olive pulp there are 39.6% tannins, that is, 39.6% of polyphenols that may be suitable for tanning. However, with the HPLC-DAD technique, there are still 46.1% of unidentified substances. To improve the identification and see if more tannins can be obtained, an identification of polyphenols was carried out by HPLC-ESI/MS that in addition to identifying more compounds the molecular mass can be determined. Therefore, it will be possible to better define the polyphenols that will have the tan capacity (i.e., polyphenols with a molecular mass between 300 and 5000 Da).

### 3.3. Polyphenol Characterization by HPLC-ESI/MS

To corroborate the tanning capacity of this raw material, the olive pulp was studied by means of the LC/MS analysis (high-resolution ESI). As mentioned before, with this technique it will be possible to refine the identification of all the polyphenols present in the olive pulp sample and at the same time, by determining their molecular mass, it will be possible to determine which of them have tanning capacity. The results obtained can be seen in Table 6. In Figure 5, the HPLC-ESI/MS chromatogram can be observed.

As can be seen in Table 6, it has been possible to detect all the polyphenols present in the sample, as well as classify them into: hydroxytyrosol and tyrosol derivatives, precursor iridoids, derivative secoiridoids, flavonoids, lignans, and phenolic acids. With the HPLC-ESI / MS technique, it was possible to detect polyphenols that had not been detected with HPLC-DAD, such as oleuropein, quercetin, hydroxypinoresinol, etc.

In Figure 6, an example of the spectrum (TIC) ESI-TOF (-) corresponding to the compound Oleuropein can be seen, that jointly to hydroxytirosol have important biological properties (anti-inflammatory, anti-atherogenic, anticancer, antimicrobial, and antiviral) as shown in many studies, and that is why it has been gaining prominence in research [22,23].

With this test it has been possible to find the molecular weights of all the polyphenols present in the sample and therefore, those that have a molecular mass between 300 and 5000 Da can be determined and will be the ones that can be used as tannins for the tanning industry. That is to say, according to the characterization carried out in this article, from the olive pulp, the following can be obtained:

-Hydroxytirosol Glucoside, Hydroxytirosol Diglucoside;-Iridoids prescursors: Loganin, Loganic acid, Secologanin, Oleoside, Oleoside Diglucoside, Oleoside Riboside, Oleoside-11-Metilester, Oleoside Dimetilester;-Secoiridoids derivatives: Oleuropein, Verbacoside, 3,4-DHPEA-EDA, p-HPEA-EA, Oleuropein Derivative 1, Oleuropein Derivative 2;-Flavonoids: Rutin, Apigenin Glucoside, Luteolin Glucoside, Quercetin;-Lignans: Hydroxypinoresinol

All of these polyphenols can be considered tannins as they have a suitable molecular weight for tanning. In addition, a tanning product with a high antioxidant activity can be obtained due to the presence of oleuropein and hydroxytyrosol that can give added value compared to the conventional vegetable extracts that are currently being used.

### 3.4. Lab Scale Application

Once it has been verified that wet olive pomace contains polyphenols that can be considered tannins, it is applied to pickled hide following the formulation shown in Table 1. In Figure 7 and Figure 8, the obtained results can be seen.

As can be seen in Figure 7 and Figure 8, leather presents a correct cross-section, and it is correctly tanned. The shrinkage temperature obtained is 79 °C. The obtained results show that the by-products obtained from olive oil extraction can be an alternative to partially replace the usual tannin extracts. However, the wet olive pomace has to be optimized in order to increase the number of tannins obtained and improve their application on leather.

## 4. Conclusions

In this work, a characterization of the polyphenols that can be obtained in two varieties of olive as well as in different by-products obtained from olive oil extraction has been carried out: wet olive pomace, degreased olive pomace, olive pulp, and olive stone, with the aim to assess the feasibility of using the by-products obtained from olive oil extraction as an alternative to totally/partially replace the usual tanning extracts and other synthetic tannin agents used today was performed.

Each type of olive and each type of fraction present a different amount of fat and tannins. Olive stone is rejected as a matter to tan leather due to the low tannin and the high non-soluble substances content.

From all the fractions studied, the fraction that contains the most number of tannins and of which more types can be identified according to the database of the equipment used is olive pulp. Specifically, in the olive pulp, there are 39.6% tannins, that is, 39,6% of polyphenols that may be suitable for tanning.

With the HPLC-ESI / MS technique, it had been possible to detect all the polyphenols present in the sample, as well as classify them into: hydroxytyrosol and tyrosol derivatives, precursor iridoids, derivative secoiridoids, flavonoids, lignans, and phenolic acids. Except for phenolic acids, most parts of these polyphenols can be considered tannins as they have a suitable molecular weight for tanning. In addition, a tanning product with a high antioxidant activity can be obtained due to the presence of oleuropein and hydroxytyrosol that can give added value compared to the conventional vegetable extracts that are currently being used.

In order to optimize the number of tannins obtained, it would be convenient to solubilize the olive pulp in order to use it as a raw material with a composition of tannins, non-tannins, and adequate fat to be able to tan and thus obtain a new bio-based resource for leather tannin.

## Figures and Tables

**Figure 1 materials-14-05790-f001:**
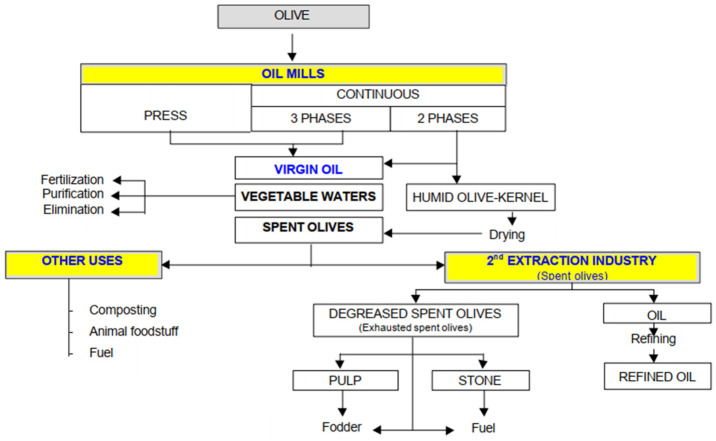
General vision on products, by-products, and residues in the olive oil industry.

**Figure 2 materials-14-05790-f002:**
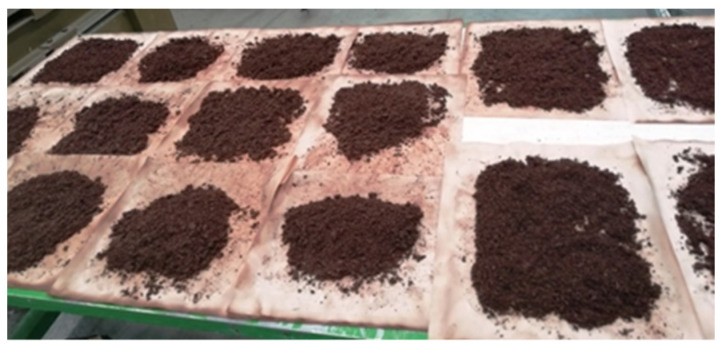
Wet olive pomace.

**Figure 3 materials-14-05790-f003:**
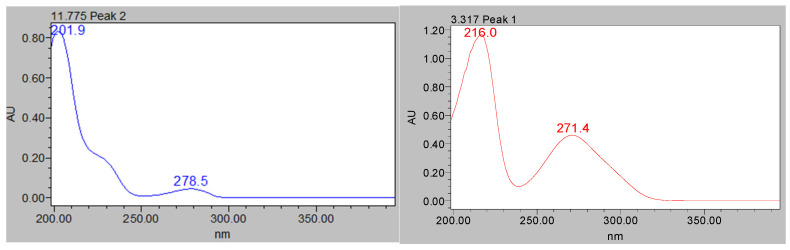
UV-Vis spectrum of catechin compounds (**left**) and gallotanines compounds (**right**).

**Figure 4 materials-14-05790-f004:**
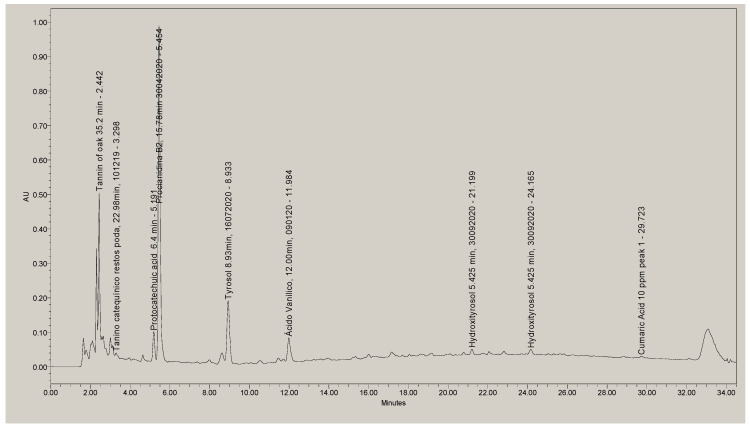
Olive pulp chromatogram.

**Figure 5 materials-14-05790-f005:**
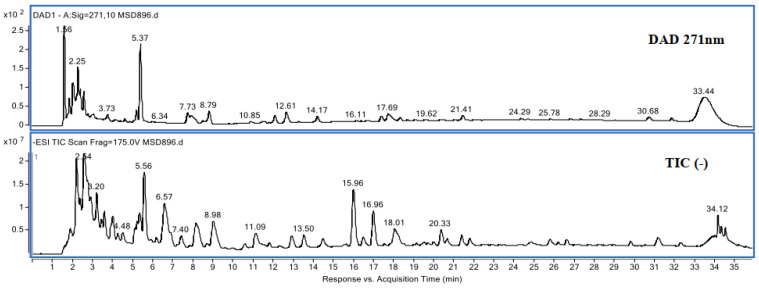
UV chromatogram (DAD) and ion chromatogram (TIC) ESI-TOF (-) Frag. 175 V.

**Figure 6 materials-14-05790-f006:**
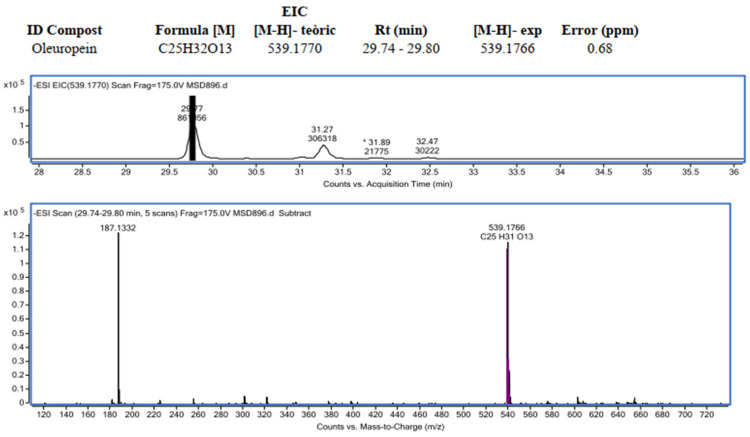
Example of the spectrum (TIC) ESI - TOF (-) of oleuropein.

**Figure 7 materials-14-05790-f007:**
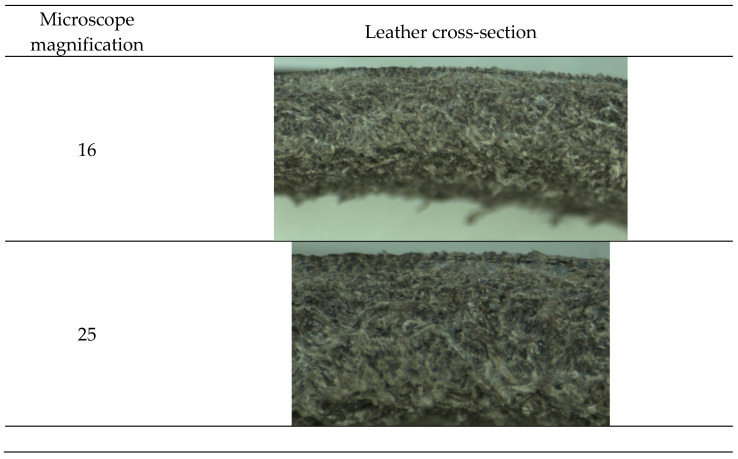
Leather cross-section.

**Figure 8 materials-14-05790-f008:**
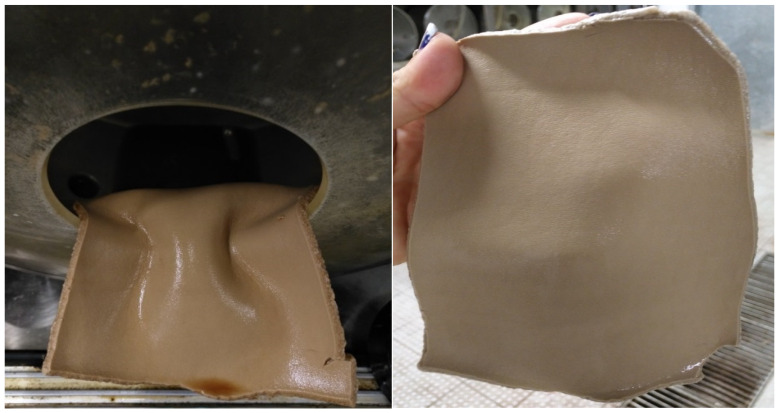
Tanned leather.

**Table 1 materials-14-05790-t001:** Formulation for WOP application.

Operation	°C	%	Product	Time (min)
Washing	20	200	Water	10
				Drain
Tanning	20	60	Water	
		0.1	EDTA	
		3.0	Disulphonic synthetic agent	
		1.0	Naftalensulphonic synthetic agent	60
		40	Wet olive pomace	
		3.0	Phenolic synthetic agent	
		1.0	Naftalensulphonic synthetic agent	
		2.0	Sulphited oil	60
		40	Wet olive pomace	
		1.0	Naftalensulphonic synthetic agent	
		3.0	Phenolic synthetic agent	Drum all night
	40	100	Water	
			Formic acid	90
			pH = 4	Drain
Washing	40	100	Water	5
				Drain
Fatliquoring	40	80	Water	
		0.2	EDTA	5
		5.0	Sulfated oil	
		1.0	Crude oil	
		0.5	Sulphited oil	90
		1.0	Disulphonic synthetic agent	30
		1.0	Formic acid	30
			pH = 3,4	Drain
Washing	20	100	Water	10
				Drain Rest 24 h.

**Table 2 materials-14-05790-t002:** Chemical characterization of the five fractions of WOP.

Sample	% Moisture	% Inorganic Matter (540 °C)	% Organic Matter (540 °C)	pH	% Fats
Palomar WOP	6.8	2.73	90.44	5.3	10.7
Arbequina WOP	10.8	2.46	86.76	5.6	7.8
Degreased WOP	6.8	6.25	86.94	5.2	3.4
Olive pulp	6.4	6.10	87.54	5.2	4.2
Olive stone	9.3	0.74	89.93	5.4	0

All the values are calculated on real weight.

**Table 3 materials-14-05790-t003:** Quantification of tannin content of the five fractions of WOP.

Sample	% No Tannins	% Soluble Substances	% Solids	% Tannins	% Non-Soluble Substances	% Moisture
Palomar WOP	2.2	4.7	33.5	2.6	28.8	66.5
Arbequina WOP	2.7	5.7	32.8	3.0	27.1	67.2
Degreased WOP	6.7	10.3	93.4	3.7	83.1	6.6
Olive pulp	12.9	16.4	93.6	3.5	77.2	6.5
Olive stone	0.3	0.9	89.9	0.7	89.9	10.1

**Table 4 materials-14-05790-t004:** Semi-quantification of tannin content of the five fractions of WOP by HPLC-DAD.

Sample	Tannins (%)	No Tannins (%)	Unidentified (%)
Palomar WOP	16.5	4.5	79.0
Arbequina WOP	31.1	16.0	52.9
Degreased WOP	13.3	10.3	76.4
Olive pulp	39.6	14.3	46.1

**Table 5 materials-14-05790-t005:** Identification of polyphenolic compounds in the olive pulp.

Peak	TR (min)	Identification	% Area/Total Area
1	1.65		2.0
2	1.79		1.7
3	2.04		1.4
4	2.10		2.5
5	2.31		5.2
6	2.44	Gallic tannin (chromophore present in oak)	10.0
7	2.58		1.3
8	2.65		2.9
9	3.01		2.5
10	3.14		0.4
11	3.30	Catechin tannin (chromophore present in quebracho)	0.7
12	3.96		0.2
13	4.18		0.3
14	4.64		0.5
15	5.19	Protocatechuic acid chromophore	2.3
16	5.45	Catechin tannin (Same chromophore as procyanidin B2)	28.9
17	7.99		0.4
18	8.62		1.6
19	8.93	Tyrosol	7.3
20	10.54		0.4
21	11.46		0.6
22	11.72		0.3
23	11.98	Vanillic acid	3.1
24	13.61		0.3
25	13.90		0.3
26	15.34		0.3
27	16.01		0.4
28	17.16		0.9
29	17.43		0.2
30	17.70		0.2
31	18.05		0.2
32	18.24		0.2
33	18.62		0.3
34	18.77		0.2
35	19.18		0.5
36	20.10		0.2
37	20.79		0.2
38	21.20	Hydroxytyrosol chromophore	0.6
39	21.74		0.2
40	22.06		0.3
41	22.81		0.3
42	24.17	Hydroxytyrosol chromophore	0.7
43	25.64		0.2
44	26.76		0.3
45	29.72	Coumaric acid chromophore	0.2
46	32.14		0.2
47	33.08		15.6
48	34.05		0.2
		Tannins	39.6
		No tannins	14.3
		Unidentified	46.1

**Table 6 materials-14-05790-t006:** Identification of polyphenolic compounds in olive pulp by HPLC-ESI/MS.

Compound	Formula	m/z	Rt	Comments
Hydroxytyrosol and Tyrosol derivatives				
Hydroxytyrosol	C_8_H_10_O_3_	153.0557	5.55	Fragments
Hydroxytyrosol Glucoside	C_14_H_20_O_8_	315.1085	5.20	Mix
Hydroxytyrosol Diglucoside	C_20_H_28_O_13_	475.1457	5.73	Fragments
Tyrosol	C_8_H_10_O_2_	137.0608	24.78	Mix
Tyrosol Glucoside	C_14_H_20_O_7_	299.1136	8.70	Mix
Iridoids precursors				
Loganin	C_17_H_26_O_10_	389.1453	13.80 17.78	Mix
Loganic acid	C_16_H_24_O_10_	375.1297	7.64 11.68	Mix
Secologanin	C_17_H_24_O_10_	387.1297	14.67 19.39 19.95	Mix
Oleoside	C_16_H_22_O_11_	389.1089	13.53	Mix
Oleoside Diglucoside	C_28_H_42_O_21_	713.2146	13.50	Mix
Oleoside Riboside	C_20_H_26_O_15_	505.1195	17.67	Mix
Oleoside-11-Metilester	C_17_H_24_O_11_	403.1246	19.10 15.86 20.28	Mix
Oleoside Dimetilester	C_18_H_26_O_11_	417.1402	18.48 21.56	Mix
Secoiridoids derivatives				
Oleuropein	C_25_H_32_O_13_	539.177	29.78 31.27	Mix
Verbacoside	C_29_H_36_O_15_	623.1981	25.96	Mix
3,4-DHPEA-EDA	C_17_H_20_O_6_	319.1187	13.98 33.81	Mix
*p*-HPEA-EA	C_19_H_22_O_7_	361.1293	18.96 19.30 30.03 33.08	Mix
Oleuropein Derivative 1	C_25_H_36_O_13_	543.2083	16.42 16.87 25.72	Mix
Oleuropein Derivative 2	C_25_H_36_O_12_	527.2134	17.07 29.02	Mix
Flavonoids				
Rutin	C_27_H_30_O_16_	609.1461	22.33	Mix
Luteolin	C_5_H_10_O_6_	285.0405	32.30	Compound
Apigenin Glucoside	C_21_H_20_O_10_	431.0984	15.33 19.07	Mix
Luteolin Glucoside	C_21_H_20_O_11_	447.0933	24.41	Mix
Quercetin	C_15_H_10_O_7_	301.0354	32.05	Mix
Lignans				
Hydroxypinoresinol	C_20_H_23_O_7_	374.1371	27.33 27.86 28.25 28.66 29.09 30.62 33.76	Mix
Phenolic acids				
Cinnamic acid	C_9_H_8_O_2_	147.0452	17.43 19.22	Mix
P-coumaric acid	C_9_H_8_O_3_	163.0401	17.75	Mix
Caffeic acid	C_9_H_8_O_4_	179.035	11.70	Mix
Vanillic acid	C_8_H_8_O_4_	167.035	4.75 12.34 20.0 23.63	Mix
Ferulic acid	C_10_H_10_O_4_	193.0506	30.10	Mix

## Data Availability

Not applicable.

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
