# Peer review of "Characterization of Wet Olive Pomace Waste as Bio Based Resource for Leather Tanning"

_materials, 2021, doi:10.3390/ma14195790_

Round 1
Reviewer 1 Report
The authors in their paper describe the “Characterization of wet olive pomace waste as bio based re-source for leather tanning”.
The work contains some degree of innovation in the intent to recycle olive oil by-products as tanning agents.
Nevertheless, no evidence of the efficiency of WOP as tanning agents is given in the paper.
Hereafter some major comments which should be addressed:
1) Information on the benefits of the use of WOP for the leather industry are totally missing and no comment is present on how these products could be introduced industrially. It seems that the paper is more a characterisation of oil by-products but in this case some information on the process of extraction and benefits/differences compared to more conventional processes should be given to improve the innovative character of the paper, which otherwise is very poor.
2) In the introduction only one sentence refers to the impact of the leather industry but no description on why this happens is given, and which possible solutions may be adopted. The authors should give experimental evidence on tanning tests carried out using WOP for the tanning of hides and comparison to commercially available tanning agents given (natural tannins and syntans).
3) At some point synthetic tannins are mentioned without any comment on why WOP could possibly substitute them.
4) A more general discussion on leather tanning should be given in the introduction and a description of different alternatives now available reported to give the reader an idea of why the leather industry has such a high environmental impact.
5) References are very poor: the authors mention that “different studies have been developed on the recovery of waste from the olive industry” but no references are given.
Even more lacking are references on the leather industry. At least references such as those listed below should be added:
- Katarzyna Chojnacka, J. Cleaner Production 2021, Volume 313, 127902
- Vanessa Gatto, Materials 2021, 14, 3069
- Oluwaseyi Omoloso, J. Cleaner Production 2020, Volume 285, 125441
- Valentina Beghetto, J. Cleaner Production 2019, Volume 220, Pages 864-872
6) What are natural tannins used for? Specify the market share. It is generally known that vegetable tannins can only be used for a very small part of the leather production and with limited applications, chemical/mechanical characteristics so how much of the WPO could be used for leather processing? Who? Which is the WPO production yearly? How much of it could be used for leather processing and which the economic advantages (if any?). some reference market data can be found at the link: https://www.euroleather.com
Author Response
The authors in their paper describe the “Characterization of wet olive pomace waste as bio based re-source for leather tanning”.
The work contains some degree of innovation in the intent to recycle olive oil by-products as tanning agents.
Nevertheless, no evidence of the efficiency of WOP as tanning agents is given in the paper.
Hereafter some major comments which should be addressed:
1) Information on the benefits of the use of WOP for the leather industry are totally missing and no comment is present on how these products could be introduced industrially. It seems that the paper is more a characterisation of oil by-products but in this case some information on the process of extraction and benefits/differences compared to more conventional processes should be given to improve the innovative character of the paper, which otherwise is very poor.
A lab scale experiment has been added in order to assess the capacity of tanning with WOP.
2) In the introduction only one sentence refers to the impact of the leather industry but no description on why this happens is given, and which possible solutions may be adopted. The authors should give experimental evidence on tanning tests carried out using WOP for the tanning of hides and comparison to commercially available tanning agents given (natural tannins and syntans).
Done.
3) At some point synthetic tannins are mentioned without any comment on why WOP could possibly substitute them.
This is a first step of a larger work that we are performing in a research project: “New recycling strategies for olive oil extraction process waste (wet olive pomace) to be applied in the leather industry” (OLIPO) MNET19/ENER-3655. MANUNET-Transnational Call 2019. This article is just the first part of the research. Once we have shown that WOP is able to tan, we follow with the research in order to improve their application and study the substitution totally or partially the conventional vegetable tanins and/or synthetic products.
4) A more general discussion on leather tanning should be given in the introduction and a description of different alternatives now available reported to give the reader an idea of why the leather industry has such a high environmental impact.
The first paragraph of the introduction has been changed.
5) References are very poor: the authors mention that “different studies have been developed on the recovery of waste from the olive industry” but no references are given.
Even more lacking are references on the leather industry. At least references such as those listed below should be added:
- Katarzyna Chojnacka, J. Cleaner Production 2021, Volume 313, 127902
- Vanessa Gatto, Materials 2021, 14, 3069
- Oluwaseyi Omoloso, J. Cleaner Production 2020, Volume 285, 125441
- Valentina Beghetto, J. Cleaner Production 2019, Volume 220, Pages 864-872
Done.
6) What are natural tannins used for? Specify the market share. It is generally known that vegetable tannins can only be used for a very small part of the leather production and with limited applications, chemical/mechanical characteristics so how much of the WPO could be used for leather processing? Who? Which is the WPO production yearly? How much of it could be used for leather processing and which the economic advantages (if any?). some reference market data can be found at the link: https://www.euroleather.com
As mentioned before, the WOP product has to be optimized, it is no closer to market. We are following our research in order to study its potential market.
Reviewer 2 Report
This paper details analysis of the tannin content of a by-product/ waste stream from the olive industry – called Wet Olive Pomace (WOP). The authors assess the tannin content and use appropriate methods to characterise individual components of the waste mixture.
I believe this paper is at an appropriate level to be published; it is well written, logically constructed, states it’s aims clearly and appropriately with a related conclusion. It was an enjoyable read and will contribute to the wider scientific pool of knowledge. There were some minor points that I think the authors should take on board / correct before publication:
- There were a few minor grammatical errors that need to be tidied e.g. a sentence mid-way through the abstract starting “In addition, leather market forecast envisages…” which is cumbersome and lacks flow.
- I strongly urge caution in suggesting that the leather industry has a high environmental impact; how would you measure this? Is water use a problem if it is fully recycled? I do appreciate why this terminology is used but great improvements have been made in processing conditions and in Europe all waste water is treated before being released or recycled and is tightly regulated and tested. Most importantly for the authors, it is not clear how this new source of veg tanning agents (from WOP) will help to improve the issues they have cited for the leather industry’s high environmental impact. Reading the opening paragraph of the introduction – which I feel paints an overly negative perspective of the leather industry – how does WOP help these figures? If it doesn’t why are they included in the introduction? My understanding is that WOP will improve the leather industries sustainability by reducing dependence on ‘virgin’ feedstocks and not because it will significantly improve the amount of water/chemical use. Some similar discussion around this can be found in a newly released book “Tanning Chemistry; the science of leather (Edition 2); RSC publishing Print ISBN: 978-1-78801-204-1.”
- The authors need to address how their leather is different to ‘wet green’ leather that is available commercially. My understanding is the commercially available olive tannages (aka wet green) are derived from the olive leaves and not WOP, but I feel it would be prudent to address this in the paper somewhere so that it is crystal clear to the reader.
- It would have been nice to see the inclusion of some small lab scale experiments on collagen to demonstrate and corroborate their findings on the tanning content. A high tannin content doesn’t always result in a good piece of leather with effective tanning.
Otherwise I am fine with the submission and would support its publication. Congratulations to the authors.
Author Response
This paper details analysis of the tannin content of a by-product/ waste stream from the olive industry – called Wet Olive Pomace (WOP). The authors assess the tannin content and use appropriate methods to characterise individual components of the waste mixture.
I believe this paper is at an appropriate level to be published; it is well written, logically constructed, states it’s aims clearly and appropriately with a related conclusion. It was an enjoyable read and will contribute to the wider scientific pool of knowledge. There were some minor points that I think the authors should take on board / correct before publication:
- There were a few minor grammatical errors that need to be tidied e.g. a sentence mid-way through the abstract starting “In addition, leather market forecast envisages…” which is cumbersome and lacks flow.
Changed.
- I strongly urge caution in suggesting that the leather industry has a high environmental impact; how would you measure this? Is water use a problem if it is fully recycled? I do appreciate why this terminology is used but great improvements have been made in processing conditions and in Europe all waste water is treated before being released or recycled and is tightly regulated and tested. Most importantly for the authors, it is not clear how this new source of veg tanning agents (from WOP) will help to improve the issues they have cited for the leather industry’s high environmental impact. Reading the opening paragraph of the introduction – which I feel paints an overly negative perspective of the leather industry – how does WOP help these figures? If it doesn’t why are they included in the introduction? My understanding is that WOP will improve the leather industries sustainability by reducing dependence on ‘virgin’ feedstocks and not because it will significantly improve the amount of water/chemical use. Some similar discussion around this can be found in a newly released book “Tanning Chemistry; the science of leather (Edition 2); RSC publishing Print ISBN: 978-1-78801-204-1.”
We appreciate this suggestion. The first paragraph of the introducion has been changed.
- The authors need to address how their leather is different to ‘wet green’ leather that is available commercially. My understanding is the commercially available olive tannages (aka wet green) are derived from the olive leaves and not WOP, but I feel it would be prudent to address this in the paper somewhere so that it is crystal clear to the reader.
Done.
- It would have been nice to see the inclusion of some small lab scale experiments on collagen to demonstrate and corroborate their findings on the tanning content. A high tannin content doesn’t always result in a good piece of leather with effective tanning.
Done.
Otherwise I am fine with the submission and would support its publication. Congratulations to the authors.
Thank you very much for your kindly revision.
Round 2
Reviewer 1 Report
Dear Authors
the required revisions and english editing have been appropriately carried out.
The manuscript in our opinion can be published in the present form.